# Management of Congenital Heart Disease: State of the Art; Part I—ACYANOTIC Heart Defects

**DOI:** 10.3390/children6030042

**Published:** 2019-03-08

**Authors:** P. Syamasundar Rao

**Affiliations:** McGovern Medical School, University of Texas-Houston, Children’s Memorial Hermann Hospital, Houston, TX 77030, USA; P.Syamasundar.Rao@uth.tmc.edu

**Keywords:** acyanotic congenital heart defects, aortic stenosis, atrial septal defect, atrioventricular septal defect, coarctation of the aorta, patent ductus arteriosus, pulmonary stenosis, ventricular septal defect

## Abstract

Since the description of surgery for patent ductus arteriosus in late 1930s, an innumerable number of advances have taken place in the management of congenital heart defects (CHDs). In this review the current status of treatment of seven of the most common acyanotic CHDs was reviewed. The discussion included indications for, and timing of, intervention and methods of intervention. The indications are, by and large, determined by the severity of the lesion. Pressure gradients in obstructive lesions and the magnitude of the shunt in left-to-right shunt lesions are used to assess the severity of the lesion. The timing of intervention is different for each lesion and largely dependent upon when the criteria for indications for intervention were met. Appropriate medical management is necessary in most patients. Trans-catheter methods are preferable in some defects while surgery is a better option in some other defects. The currently available medical, trans-catheter, and surgical methods to treat acyanotic CHD are feasible, safe, and effective.

## 1. Introduction

Since the description of ductal ligation by Gross and Hubbard [1] in the late 1930s, utilizing digoxin to treat children with congenital heart defects (CHDs) by Gibson [2] and Taussig [3] in mid-1940s, and trans-catheter treatment of tricuspid and pulmonary valve stenosis by Rubio and Limon-Lason [4,5] in the early 1950s, remarkable advances in the management of CHD have occurred. The author has, from time to time, reviewed the diagnostic and management aspects of CHD [6,7,8,9,10,11,12,13,14,15,16]. In this paper, the author reviews the current state of the art management aspects of the most common CHD. For each CHD, a brief description of the defect will be included followed by indications for intervention, timing of intervention, and type of intervention. Due to the extensive nature of the material, discussion of acyanotic CHD will be included in this paper and cyanotic CHD will be discussed in the next paper.

## 2. Pulmonary Stenosis

Pulmonary stenosis may be at valvar, subvalvar, or supravalvar levels or in the branch pulmonary arteries (PAs). Valvar stenosis is the most common type of right ventricular (RV) outflow tract obstruction and constitutes 7.5% to 9.0% of all CHDs [7,12]. The pathology of valvar pulmonary stenosis is a “dome-shaped” pulmonary valve with fusion of pulmonary valve leaflets protruding into the pulmonary artery as a conical, windsock-like structure. Hypertrophy of the right ventricle in proportion to the degree of obstruction and dilatation of the main PA independent of the degree of obstruction are seen.

Treatment of valvar pulmonary stenosis is included in this section, but discussion of other RV outflow tract obstructions, such as infundibular stenosis, double-chamber RV, sub-pulmonary stenosis in patients with corrected transposition of the great arteries, and supra-valvar and branch PA stenosis will not be included because of limitations of space, but the interested reader is referred to the author’s prior publication [17].

### 2.1. Indications for Intervention

Management of pulmonary stenosis is largely dependent upon the severity of the obstruction. The natural history study [18] is helpful in categorizing the degree of narrowing: trivial—peak-to peak gradients across the pulmonary valve ≤25 mmHg; mild—gradients between 25 and 49 mmHg; moderate—50 to 79 mmHg; and severe—≥80 mmHg. In patients with trivial and mild pulmonary valve stenosis, periodic clinical follow-up, antibiotic prophylaxis prior to any bacteremia-producing procedures to prevent subacute bacterial endocarditis (SBE), and no exercise restriction are recommended. Indications for intervention are peak-to-peak gradients in excess of 50 mmHg [19]. Some cardiologists advocate balloon valvuloplasty for lesser gradients, but the author believes that the indications for intervention should remain the same as for surgical intervention and the reasons for such recommendation were detailed in a prior review for the interested reader [19].

### 2.2. Timing of Intervention

The timing for intervention is dependent upon the degree of obstruction. In children, when the gradient reaches 50 mmHg, elective intervention should be entertained. Urgent intervention may be necessary in the neonates and young infants with critical obstruction, defined as supra-systemic right ventricular systolic pressure, right to left inter-atrial shunt, and/or ductal dependent pulmonary circulation [20,21].

### 2.3. Type of Intervention

Currently, balloon pulmonary valvuloplasty is the treatment of choice for valvular pulmonary stenosis. A detailed description of the procedure of balloon valvuloplasty is beyond the scope of this paper, but can be found elsewhere [22]. In brief, a balloon valvuloplasty catheter is positioned across the pulmonary valve and the balloon is inflated (Figure 1); the radial forces of balloon inflation produce disruption of the valve leaflet fusion causing relief of the pulmonary valve obstruction. While the initial recommendations [23] were to use balloons that are 1.2 to 1.4 times the size of the pulmonary valve annulus, more recent recommendations are to limit the balloon/annulus ratio to 1.2 to 1.25 [22,24]. In most cases, single balloons are used for balloon pulmonary valvuloplasty (Figure 1). If the pulmonary valve annulus is too large to dilate with a single balloon, two balloons placed across the pulmonary valve were employed to accomplish the procedure prior to the availability of large-diameter balloons. Now that larger balloons are commercially available, the need for use of the double balloon technique has diminished. When single and double balloon techniques were compared, there was no advantage for the use of the double balloon technique over the conventional single balloon technique [25,26]. Since balloon pulmonary valvuloplasty is successful in most cases (Figure 2) [22,27], surgical management is reserved for cases with supravalvular pulmonary artery stenosis, severe valve annular hypoplasia, and dysplastic pulmonary valves.

Development of pulmonary insufficiency following balloon pulmonary valvuloplasty at late follow-up is well documented [24,27]; some of these patients may requires surgical or trans-catheter pulmonary valve replacement in the future.

## 3. Aortic Stenosis

Aortic stenosis may be at valvar, subvalvar (fixed subaortic stenosis and idiopathic hypertrophic subaortic stenosis), and supravalvar sites [28]. Valvar stenosis is the most common form of left ventricular (LV) outflow tract obstruction; the prevalence is 5–6% of patients with CHD [7,12]. The anatomy of the stenotic aortic valve is variable; the bicuspid valve is most common with varying degrees of commissural fusion. The valve leaflets are thickened, domed, and non-pliable. Tricuspid and rarely unicuspid aortic valve leaflets may be responsible for aortic valve obstruction. Aortic valve leaflet dysplasia with or without hypoplasia of the valve annulus and unicuspid aortic valve may be seen in neonates and young infants. Calcification of the aortic valve leaflets, frequently seen in the elderly, is uncommon during childhood. Concentric hypertrophy of the LV muscle is present and is largely proportional to the degree of obstruction. Dilatation of ascending aorta (post-stenotic dilatation) is seen in most cases, but the degree of aortic dilatation is independent of the severity of aortic stenosis [12,28].

Discussion of valvar aortic stenosis is included in this section, but management of subvalvar (fixed subaortic stenosis and idiopathic hypertrophic subaortic stenosis) and supravalvar aortic stenosis is not dealt with here because of limitations of space, but can be found elsewhere [28].

### 3.1. Indications for Intervention

Management of aortic valve stenosis is also based upon the severity of obstruction, similar to pulmonary stenosis. In patients with trivial and mild aortic valve stenosis, periodic clinical follow-up, antibiotic prophylaxis prior to any bacteremia-producing procedures to prevent SBE and no exercise restriction are suggested. The indications for intervention in valvar aortic stenosis is a peak-to-peak gradient in excess of 50 mmHg with either symptoms or electrocardiographic (ECG) ST-T wave changes or a peak gradient more than 70 mmHg irrespective of symptoms or ECG changes [28,29].

### 3.2. Timing of Intervention

Elective intervention is indicated when criteria alluded to above are met. Emergent intervention is indicated in the neonates and young infants with a high gradient, congestive heart failure (CHF), or ductal dependent systemic circulation.

### 3.3. Type of Intervention

In the past, surgical commissurotomy was the treatment of choice. Since the introduction of balloon valvuloplasty for valvar aortic stenosis in 1983, it has been used as a first therapeutic procedure for relief of aortic valve obstruction. Detailed description of the procedure of balloon valvuloplasty is beyond the scope of this paper, but is described elsewhere [29,30,31]. In brief, a balloon valvuloplasty catheter is positioned across the aortic valve and the balloon is inflated (Figure 3); the radial forces of balloon inflation produce disruption of valve leaflet fusion causing relief of aortic valve obstruction. The recommendations were to use a balloon diameter size 80–100% of the aortic valve annulus. The considerations for use of the single vs. double balloon technique is similar to those alluded to in the pulmonary stenosis section. Sometimes, it is difficult to maintain the balloon across the aortic valve during balloon aortic valvuloplasty. In such situation, use of extra-stiff wires, longer balloons, double balloon technique, adenosine administration to produce cardiac standstill, and overdrive pacing have been used to accomplish a successful procedure [29,30,31]. In the author’s personal experience use of extra-stiff wires and longer balloons was helpful in carrying out the procedure with occasional need for overdrive pacing. Since balloon aortic valvuloplasty is successful in most cases (Figure 4) [29,30,31,32], surgery is reserved for unsuccessful balloon cases and to address complications of balloon aortic valvuloplasty. Neonates with unicuspid aortic valves are also candidates for surgery. While many centers are exhibiting enthusiasm for trans-catheter aortic valve replacement (TAVR), it should be emphasized that the TAVR should be reserved for calcific aortic stenosis of the elderly and the non-calcific aortic stenosis in adolescents and adults should be addressed by the less invasive, simpler balloon procedure [33].

Recurrence of aortic stenosis and development of aortic insufficiency after balloon aortic valvuloplasty at late follow-up is well documented [31,32,33]. Repeat balloon aortic valvuloplasty, surgical valvotomy, or valve replacement may be required in the future in some of these patients to adequately address residual aortic stenosis and/or insufficiency.

## 4. Coarctation of the Aorta

Coarctation of the aorta (CoA) is a congenital cardiac anomaly with obstruction to blood flow in the descending aorta. The prevalence CoA varies between 5% and 8% of all CHDs [7,12]. Classic CoA is located in the thoracic aorta distal to the origin of the left subclavian artery, at about the level of the ductal structure and comprises of narrowed aortic segment with localized medial thickening and infolding of the media and superimposed neointimal tissue [34,35,36]. A shelf-like structure or a membranous curtain-like structure may be seen with either an eccentric or a central opening. It may be discrete or a long segment of the aorta may be involved. It may be present as an isolated lesion or may be seen in association with other CHD, particularly in the neonate [34]. Rarely, the coarcted segment may be present in the lower thoracic or abdominal aorta. These lesions may be long and fusiform in nature with irregular lumen and are likely to be variants of Takayasu arteritis. Varying degrees of LV hypertrophy are seen with aortic coarctation. Collateral vessels develop in older children.

### 4.1. Indications for Intervention

Indications for intervention, either by surgery or balloon angioplasty are significant hypertension and/or CHF with a peak-to-peak systolic pressure gradient in excess of 20 mmHg across CoA [34,35,36,37].

### 4.2. Timing of Intervention

Elective intervention is recommended for infants, children, adolescents, and adults who are asymptomatic. In the absence of hypertension or CHF, elective therapy in children between the ages of one and five years was originally suggested, but more recent data indicate that intervention should be undertaken prior to one year of age in order to prevent hypertension later in life. Neonates and infants who are symptomatic should have urgent intervention as soon as the baby is stabilized.

### 4.3. Type of Intervention

At the present time, we have surgery, balloon angioplasty (Figure 5) and stents (Figure 6), both bare metal and covered, for treating CoA. There are no uniformly agreed criteria to decide on which therapy is best for a given clinical scenario, but generally used treatment algorithms depend upon the age of the patient at the time of presentation and the anatomy of the coarctation (discrete vs. long segment) [37].

Children older than six months with discrete native coarctation should be treated with balloon angioplasty since the results are generally good at this age (Figure 7). Patients with long-segment coarctations and those associated with significant isthmic hypoplasia may have surgical correction. However, older children, adolescents, and adults may be treated with stent implantation.

Neonates and infants (<6 months) with CoA are treated by surgical intervention because of high recurrence rates following balloon angioplasty, particularly in the neonate. Surgical options include resection and end-to-end anastomosis, subclavian flap angioplasty, prosthetic patch angioplasty, tubular bypass grafts, and variations thereof. Type of surgery used is in large part dependent upon the anatomy of the CoA and to some extent by surgeon’s preference. However, balloon angioplasty in neonates and young infants is useful in critically ill infants, particularly in those in whom averting aortic cross-clamping required for surgery is valuable in the overall management. Such special situations are infants with shock-like syndrome [38], severe myocardial dysfunction and hypertensive cardiomyopathy [39], prior spontaneous cerebral hemorrhage [40], severely compromised left ventricular function [39,40], and biliary atresia awaiting liver transplantation [40].

Development of recoarctation following surgery is independent of the type of surgical repair in that it has been seen following all types of surgery described above [41,42]. Recurrent coarctation after previous surgery is generally treated with balloon angioplasty [41,42,43]. Recoarctation following prior balloon angioplasty may be treated with repeat balloon angioplasty [44]. If the recoarcted segment is long and/or is associated with hypoplastic isthmus, surgery in young children and stents in older children and adults is most appropriate.

A detailed description of the balloon angioplasty procedure is beyond the scope of this paper, but is described elsewhere [35,37,41]. In brief, a balloon angioplasty catheter is positioned across the coarctaion segment and the balloon is inflated (Figure 5); the radial forces of balloon inflation produce disruption of the obstructive tissue causing relief of aortic obstruction. The size of the balloon selected for angioplasty should be two or more times the size of the coarcted segment, but no larger than the size of the descending aorta at the level of the diaphragm [34,35,37,41]. Special techniques, such as those alluded to in the aortic stenosis section are not necessary since the balloon position across the coarctation segment is generally stable.

Adolescents and adults irrespective of the type of coarctation (native, postsurgical, or post-balloon) should be treated with stent implantation. Stents are not generally recommended for young children [45,46,47] because the stents do not grow as the child grows and may necessitate a more complicated surgical repair at a later date. However, wider availability of stents, such as the Valeo Biliary Pre-mounted Re-dilatable Stent (Edwards Life Sciences, Irvine, CA, USA) [48], which can be expanded up to 20 mm, may obviate such a problem.

Covered stents may be used in very special circumstances such as sub-atretic type of coarctations and those associated postsurgical or post-balloon aneurysms [49,50].

Management of native and post-surgical coarctations was included above, but treatment of Takayasu arteritis was not discussed because of limitations of space, but can be found elsewhere [50].

## 5. Atrial Septal Defect

Five major types of atrial septal defects (ASDs) are recognized; these are ostium secundum, ostium primum, sinus venosus, and coronary sinus defects and patent foramen ovale (PFO) [51,52]. Each of these defects will be addressed separately.

### 5.1. Ostium Secundum ASDs

In ostium secundum ASDs, there is deficiency of the atrial septal tissue in the region of the fossa ovalis. ASDs constitute 8–13% of all CHDs [7,12]. These defects may be small, medium, or large in size; their shape may be oval, circular, or irregular. Most of the ASDs are single defects; however, multiple defects and fenestrated defects may occasionally be seen. Because of left-to-right shunt across the ASD, the right atrium and right ventricle are dilated and somewhat hypertrophied. Likewise, main and branch PAs are also dilated. Pulmonary vascular obstructive disease (PVOD) does not develop until adulthood.

#### 5.1.1. Indications for Intervention

Small ASDs (<5 mm) are likely to spontaneously close and do not need closure. Moderate to large defects may require occlusion. Indications for closure of the ASDs are RV volume overloading (dilatation of the right heart with flat to paradoxical inter-ventricular septal motion) on echocardiogram and/or a pulmonary-to-systemic flow ratio (Qp:Qs) greater than 1.5:1 [51,52]. Even in the absence of symptoms at presentation, closure of moderate to large ASDs is recommended in order to prevent development of PVOD in adulthood, decrease the probability of supra-ventricular arrhythmias later in life, and to prevent development of symptoms during adolescence and adulthood.

#### 5.1.2. Timing of Intervention

Elective closure around four to five years of age is recommended. Closure during infancy is not undertaken unless the infant is symptomatic [15,51]. ASDs in adults should be closed at presentation, electively, irrespective of their age and symptoms [52,53]. The reasons for such recommendation, as reviewed in detail elsewhere [52,53], are that untreated adult ASD patients have decreased event-free survival rates when compared to the normal population, closure by surgery is safe and effective with high event-free survival rates, and ASD closure prevents functional deterioration, improves cardiac function, and increases functional capacity [52,53]. Additionally, the earlier the closure, the better the long-term outlook. Therefore, it is recommended that closure of hemodynamically significant ASDs in all adults be undertaken as soon as they are identified [52,53].

#### 5.1.3. Type of Intervention

Surgical and trans-catheter closure are the currently available options for addressing ASDs. While surgical closure of ASDs is safe and effective with low mortality, the morbidity associated with sternotomy/thoracotomy cannot be avoided. Since the initial description by King [54], Rashkind [55], and their associates, a number of ASD occlusion devices were described, as reviewed elsewhere [56,57,58,59]. However, at the present time, only the Amplatzer^®^ Septal Occluder (ASO) (St. Jude Medical, Golden Valley MN, USA), Amplatzer^®^ Cribriform device, and Gore HELEX^®^ devices (W.L. Gore, Flagstaff, AZ, USA) are approved for clinical use by the FDA in the USA. The Occlutech Septal Occluder (Occlutech International, Helsingborg, Sweden) with design features similar to the ASO is used outside the USA. Studies comparing surgical vs. device closure suggest similar effectiveness, but device closure is less invasive, requires no cardiopulmonary bypass, has fewer complications (10 vs. 31%), requires shorter hospital stays (1 vs. 4.3 days), and is less expensive ($11,000 vs. $21,000) [60,61,62]. The device closure techniques have proven to be safe, cost-effective and favorably compare with surgical closure [60,61,62,63]. Trans-catheter closure of ASDs using various devices is now an established practice in most cardiac centers [63,64].

A detailed description of the procedure trans-catheter occlusion of ASDs with Amplatzer^®^ Septal Occluder (ASO) (St. Jude Medical, Golden Valley MN, USA), Amplatzer^®^ Cribriform device (St. Jude Medical, Golden Valley MN, USA), and Gore HELEX^®^ devices (W.L. Gore, Flagstaff, AZ, USA) is beyond the scope of this paper, but is described elsewhere [15,51,52,53]. In brief, the left atrial component of the device is delivered into the left atrium, waist of the device in the defect and the right atrial component into the right atrium, all under fluoroscopic and trans-esophageal (TEE) guidance (Figure 8, Figure 9 and Figure 10). Trans-catheter occlusion of ASDs is successful in most cases. At the present time, surgery is mostly reserved for ASDs with poor septal rims deemed difficult to close with percutaneous techniques or unsuccessful attempts to close the defect by trans-catheter techniques. Also, if the repair of other associated defects is anticipated, surgical closure of the ASD may be performed concurrently during that surgery.

### 5.2. Ostium Primum ASDs

Ostium primum ASDs belong to the spectrum of CHD called atrio-ventricular septal defects (AVSDs) and were previously known as endocardial cushion defects and atrio-ventricular canals. These ASDs are usually large in size and are located in the anterior portion of the lower part of the inter-atrial septum. A cleft in the anterior leaflet of the mitral valve is almost always present and results in varying degrees of mitral insufficiency. A cleft in the septal leaflet of the tricuspid valve may be seen in some patients. The left ventricular outflow tract is long and narrow and is described as a goose-neck deformity. Sometimes left ventricular outflow tract obstruction may be present. Right heart structures are dilated in a manner similar to that described for ostium secundum ASDs. Left atrium (LA) and LV may also be dilated if mitral insufficiency is moderate to severe in degree.

#### 5.2.1. Indications for Intervention

CHF is rare with ostium primum ASDs, and when present, it is seen in association with severe mitral insufficiency. In these patients anti-congestive measures (diuretics, afterload-reducing agents, and digoxin) should be promptly instituted. The indications for repair are similar to those described above for the closure of ostium secundum ASDs. The presence of CHF and significant mitral insufficiency call for intervention irrespective of the magnitude of the left-to-right shunt across the ASD component of the defect. SBE prophylaxis is recommended and normal activity is allowed in the absence of severe mitral insufficiency.

#### 5.2.2. Timing of Intervention

While surgical correction can be performed at any age, surgical repair in asymptomatic patients is usually recommended between the ages of three and five years [15,51]. In symptomatic patients, or in patients with severe mitral insufficiency, surgical repair may be performed at presentation following medical control of CHF. Some surgeons advocate surgical correction of asymptomatic ostium primum ASDs in infancy [65,66,67], but careful review of the results indicated higher mortality rates in the infant group [67] without accruing long-term benefit [68]. Consequently, it is appropriate to perform surgical repair between the ages of three and five years for the asymptomatic patient with ostium primum ASD.

#### 5.2.3. Type of Intervention

Trans-catheter occlusion, now the usual treatment for ostium secundum ASDs, is not possible for ostium primum ASDs because there are no inferior septal rims, but more notably because of the need for addressing mitral valve cleft and consequent mitral insufficiency. The standard treatment of choice of ostium primum ASDs is surgical correction; the atrial defect is closed using an autologous pericardial patch and closure of the mitral valve cleft with interrupted suture material is performed under cardiopulmonary bypass.

### 5.3. Sinus Venosus ASDs

Sinus venosus ASDs are of two types; the majority of defects are located in the posterior superior portion of the inter-atrial septum, frequently overriding the orifice of the superior vena cava. Occasionally, the ASD may be situated in the inferior-posterior part of the atrial septum, overriding the inferior vena cava. The sinus venosus ASDs are frequently associated with anomalous pulmonary venous connection; the right upper pulmonary veins and rarely the veins from the entire right lung empty into the superior vena cava or to the right atrium near the cavo-atrial junction. Sinus venosus ASDs constitute 5–10% of all ASDs [51]. Right heart dilatation is similar that described for ostium secundum ASDs.

The indications for intervention are also same as those for the ostium secundum ASDs. However, these defects are not amenable to trans-catheter closure because of the lack of superior rim and the need for redirecting the anomalous pulmonary vein(s) into the left atrium. However, some attempts to avoid such problems by the implanting a covered stent to direct the superior vena caval flow into the right atrium while occluding the atrial defect and redirect the anomalous pulmonary vein into the left atrium have been made, as mentioned elsewhere [15]. Currently, surgical correction is the treatment of choice. Diversion of the anomalously connected right pulmonary vein(s) into the left atrium along with the closure of the ASD are performed under cardiopulmonary bypass. This may involve constructing a tunnel with an autologous pericardial patch along with enlargement of superior vena cava. The results are usually good, with rare superior vena caval or pulmonary venous obstruction.

### 5.4. Coronary Sinus ASDs

Coronary sinus ASDs are rare [51], perhaps are less than 1% of all ASDs. These are defects in the inferior and anterior part of the atrial septum at the expected site of the opening of the coronary sinus. These defects are frequently related to a persistent left superior vena cava and un-roofing of the coronary sinus, a complex initially described as Raghib syndrome. Dilatation of right heart structures is similar to that described for ostium secundum ASD.

The indications for intervention are also the same as those for the ostium secundum ASDs, namely volume overloading of the right heart. Surgical correction with patch closure of the defect, leaving the entry of coronary sinus in the left atrium is the standard method of approach; this is performed under cardiopulmonary bypass. Results of surgery are generally good. These defects are not usually amenable to trans-catheter occlusion; however, small defects may be amenable to percutaneous occlusion [69].

### 5.5. Patent Foramen Ovale

The final defect in the atrial septum to be discussed is PFO. In the fetus the foramen ovale is kept patent because of the mechanical effect of streaming of the inferior vena caval blood into the left atrium [70,71,72]. Following birth, a combination of increased left atrial pressure secondary to increased pulmonary venous return and a fall in the right atrial pressure due to elimination of placental return produces apposition of the septum primum and septum secundum with resultant functional closure of the foramen ovale [70,71,72]. Ultimately, anatomical closure occurs in most normal individuals. However, autopsy studies in the past and more recent TEE studies indicate that PFO is present in nearly one-third of the normal population and therefore, PFO should be considered a normal variant.

The left-to-right shunt across the PFO is usually trivial and not problematic and no treatment is necessary in otherwise normal hearts. However, the PFOs become significant in the presence of other CHDs or if they are the source of right to left shunt producing paradoxical embolism with resultant cerebrovascular accidents (CVAs)/transient ischemic attacks (TIAs) or other problems, such as migraine, Caisson’s disease, and platypnea-orthodeoxia syndrome.

#### 5.5.1. PFOs Associated with Other CHD

In neonates with right heart obstructive lesions such as tricuspid or pulmonary atresia and left heart obstructive lesions such as hypoplastic left heart syndrome and mitral or aortic atresia, continued patency of the foramen ovale is critical so that an obligatory right-to-left or left-to-right shunt, respectively, across the atrial septum can take place. In a similar manner, patency of the foramen ovale is important in total anomalous pulmonary venous connection where all pulmonary and systemic venous returns come into the right heart and consequently the systemic blood flow is completely derived from right-to-left shunting across the PFO. In newborn babies with transposition of the great arteries (TGA), the circulation is parallel (instead of normal in-series circulation) and mixing across the circulations is essential for survival; this is usually provided by the ASD/PFO. With any of the above scenarios the foramen ovale can become restrictive and may need enlargement either by trans-catheter or surgical techniques [73]. Apart from palliation with enlargement of PFO, primary cardiac problem should be addressed and these issues will be discussed in Part II.

#### 5.5.2. Residual PFOs in Previously Treated Complex CHD

Arterial desaturation may be present because of right to left shunt via PFO in patients who were previously treated for complex congenital cardiac anomalies, including Fontan fenestrations. These defects should be closed, preferably with a trans-catheter methodology [74].

#### 5.5.3. PFOs Presumed to be the Seat of Paradoxical Embolism

Paradoxical embolism through a PFO may be an important cause of CVA/TIA in young adults. Closure of these PFOs is generally recommended to prevent recurrence of paradoxical embolism and CVA/TIA. While there is no consensus, most physicians consider trans-catheter closure of PFO [75] is preferable to life-long anticoagulation and surgical closure.

#### 5.5.4. PFOs in Platypnea-Orthodeoxia Syndrome

Dyspnea and arterial desaturation in upright position is known as platypnea-orthodeoxia syndrome [76,77,78]. It is a rare condition seen in elderly subjects and has incapacitating symptoms and appear to be secondary to increase in right-to-left shunt across ASD/PFO upon assuming upright position. Both surgical and trans-catheter methods were used in the past to occlude these ASDs/PFOs.

#### 5.5.5. PFOs in Other Conditions

Symptoms in patients with right ventricular infarction, decompression (Caisson’s) illness and migraine have also been attributed to right to left shunt across the PFO. There is varying degrees of evidence regarding the benefits of occlusion of PFOs in above described conditions.

### 5.6. Type of Intervention

Surgical and trans-catheter closure are the currently available options to address these PFOs. Surgical closure can be performed in most of the above PFOs and may be preferable if other surgical interventions are necessary concurrently. Most cardiologists prefer trans-catheter closure. Devices used are essentially those that are used for ASD closure; however, the Amplatzer PFO occluder (St. Jude Medical, Inc., Golden Valley, MN, USA) is suitable to close these PFOs given the FDA approval of the Amplatzer PFO occluder. Other FDA-approved devices to close PFOs for prevention of recurrence of stroke are the Gore Cardioform Septal Occluder (W.L. Gore and Associates, Flagstaff, AZ, USA), Amplatzer, and Gore HELEX^®^ devices (W.L. Gore and Associates, Flagstaff, AZ, USA).

## 6. Ventricular Septal Defect (VSD)

Defects in the ventricular septum are generally classified on the basis of their location in the inter-ventricular septum and are divided into perimembranous (situated in the membranous ventricular septum in the subaortic region), supracristal (located in the conal septum in the subpulmonary region), atrioventricular (AV) septal (positioned in the inlet septum), and muscular (sited in the muscular and apical areas of the ventricular septum). VSDs constitute 20–25% of all CHDs [7,12]. The perimembranous VSDs are the most common defects and constitute 80% of the VSDs. The prevalence of supracristal (5–7%), AV septal (8%), and muscular (5–20%) types of VSDs is less common. The VSDs may be large, medium, or small in size. The majority of the defects are single. Infrequently, multiple defects are present in the muscular septum, described as the “Swiss-cheese” VSDs. A left-to-right shunt across the VSD produces dilatation of the LA and left LV. The magnitude of the shunt is largely dependent upon the effective diameter of the VSD. The RV and main and branch PAs are dilated in moderate to large VSDs. While PVOD does not manifest until adulthood in patients with ASD, patients with VSD can develop PVOD as early as 18 months to two years of age if a large VSD is not repaired.

VSDs seen in association with tetralogy of Fallot, pulmonary atresia/stenosis, transposition of the great arteries, tricuspid and mitral atresia, and double-outlet RV and heterotaxy (asplenia and polysplenia) syndromes will not be addressed in this paper, but will included in Part II. Likewise, post-traumatic and post-myocardial infarction VSDs will not be discussed in this review.

### 6.1. Indications for Intervention

The indications for intervention are largely based on the size and type (location) of the VSD. There is great natural tendency for spontaneous closure of the VSD and this should be kept in mind when considering treatment of VSDs.

*Small VSDs* do not require closure; assurance of the parents and perhaps SBE prophylaxis and infrequent clinical follow-up are recommended. However, if the VSD is small because of its partial closure by prolapsed aortic valve cusp into the defect, causing aortic insufficiency, surgical closure of the defect with re-suspension of the aortic valve leaflets is appropriate.

*Moderate VSDs* with CHF should first be treated with anti-congestive measures. If failure to thrive, markedly enlarged LA and LV, or elevated PA pressures (or both) are present, closure of the defect is performed. The criterion for closure is a Qp:Qs greater than 2:1.

*Large VSDs* with RV and PA systolic pressures similar to LV and aortic systolic pressures should be closed.

*VSDs with elevated pulmonary vascular resistance (PVR)* may be closed if the calculated PVR index is less than six Wood units or the pulmonary-to-systemic vascular resistance ratio (Rp:Rs) is less than 0.35 (or both) with a Qp:Qs greater than 1.5. If the PVR is higher, pulmonary vascular reactivity testing with oxygen and nitric oxide (NO) should be undertaken. If the PVR index decreases to values below 6–8 units with oxygen or NO, VSD closure is indicated. Patients with large VSD with irreversible PVOD are not candidates for VSD closure. Pulmonary vasodilators such as Sildenafil and Bosantin may be tried. These patients may ultimately be considered for lung transplantation.

### 6.2. Timing of Intervention

VSD closure should be performed prior to 6–12 months of age (certainly no later than 18 months of age) irrespective of control of heart failure and adequacy of weight gain. The purpose is to prevent irreversible PVOD. In infants with Down syndrome, VSD closure should be performed prior to six months of age because these patients tend to develop PVOD earlier than non-Down patients.

### 6.3. Type of Intervention

The available treatment options for VSDs are medical, trans-catheter, surgical, and hybrid approaches. The clinical status of the patient, size, and location of the VSD determine the method selected for a given patient.

#### 6.3.1. CHF

Infants with moderate to large VSDs may develop CHF; these babies should receive prompt treatment with anti-congestive measures to include diuretics (furosemide and aldactone), and afterload reducing agents (angiotensin-converting enzyme inhibitors: captopril/enalopril and others), and digoxin, as appropriate. Overall management includes optimization of nutrition, maintenance of adequate hemoglobin level, and addressing the associated respiratory symptoms [8,16].

#### 6.3.2. Perimembranous VSDs

Surgical closure under cardiopulmonary bypass remains the main treatment option for large and non-restrictive perimembranous VSDs. Most of the perimembranous VSDs are closed with a Dacron patch via right atriatomy with or without detachment of the tricuspid valve leaflets. Surgical closure of VSDs is safe and effective with low mortality rates (less than 3%). A common practice in an earlier era of initial pulmonary artery banding followed by surgical closure of the VSD later is no longer done, instead, primary surgical correction is undertaken at the present time. However, such a staged approach may be used for the Swiss-cheese variety of muscular VSDs.

A number of devices have been used in percutaneous closure of the VSDs, as reviewed elsewhere [16]. Most of the devices are double-disc devices, requiring septal rims to hold the device in place. To address this problem, Amplatzer Membranous VSD Occluder (St. Jude Medical, Inc., Golden Vally MN, USA) was designed and clinical trials were performed including FDA-approved US clinical trials [79]. Although the results were generally considered acceptable, development of complete heart block [16,80] both immediately after and at follow-up in 1 and 22% of patients was of concern. This is in contradistinction to 1% incidence of complete heart block after surgical closure. Consequently, the Amplatzer Membranous VSD occluders (St. Jude Medical, Inc., Golden Vally MN, USA) are no longer used [16] nor did the FDA approve the device.

#### 6.3.3. Supracristal VSDs

Supracristal defects are usually closed by surgery via the pulmonary valve. VSDs with associated aortic insufficiency, even though small, should be closed to prevent progression of aortic insufficiency. Re-suspension of the aortic valve or other valvuloplasty techniques may have to be used in patients with moderate to severe aortic valve prolapse.

#### 6.3.4. AV Septal VSDs

The discussion of AV septal VSDs will be included in the AVSD section below.

#### 6.3.5. Muscular VSDs

Small muscular VSDs are likely to close spontaneously and do not require intervention. Moderate to large muscular VSDs should be closed. Several different approaches are currently available and include surgical closure, initial pulmonary artery banding with later surgical closure, percutaneous closure and hybrid approach.

Surgical Closure under cardiopulmonary bypass is feasible, but because of heavy ventricular septal trabecuations and multiple VSD opening on the right side, significant residual shunts are seen following surgery. Left ventriculotomy may address this problem, but left ventriculotomy in early infancy is not well tolerated.

Pulmonary artery banding initially to control CHF, reduce pulmonary blood flow and decrease pulmonary artery pressure is most commonly used in babies less than three months of age. The VSD is closed via an apical left ventriculotomy later during childhood. The PA band is removed and the PA is repaired at that time, as necessary. Sometimes the muscular VSDs close spontaneously after PA banding, and repeat surgery to remove the pulmonary band is required. An absorbable (polydioxanone) PA band decreases pulmonary blood flow and pressure initially and helps abate symptoms of CHF. As the VSD closes spontaneously, the PA band is resorbed and does not require repeat surgery to remove it. The philosophy is similar to those advocated for patients with tricuspid atresia with a large VSD [81].

Percutaneous closure seems to be a valuable option for closure of large muscular VSDs. As mentioned above, a number of devices were developed to percutaneously close of the VSDs [16]; Amplatzer Muscular VSD Occluder (St. Jude Medical, Inc., Golden Valley, MN, USA) stood the test of time and was approved for clinical use by the US FDA. The method of implantation is described elsewhere for the interested reader [16]. Results of muscular VSD closure with Amplatzer Muscular VSD Occluder are generally good with complete closure rates of 70% at six months and 92% at 12 months after the procedure [82]. However, multiple and serious complications, particularly in the small infants, have been reported as reviewed elsewhere [16].

*Hybrid Device Closure.* Since there are inherent limitations to both surgical and trans-catheter device closure of muscular VSDs in small babies, a hybrid procedure, also described as “perventricular closure”, was initiated [83,84]. Several other cardiologists adopted this technique subsequently [16]. Detailed description of the technique was reviewed elsewhere for the interested reader [16,83,84]. The results are excellent with high closure rates (89–96%) [85,86]; the few unsuccessful procedures were changed to conventional open-heart surgical repair, irrespective of the reason for failure. This hybrid procedure is particularly helpful in small babies with large muscular VSDs in whom a higher prevalence of adverse events for percutaneous device closure exists [16].

#### 6.3.6. Comments

Careful examination of the data of patients undergoing VSD closure reveal that many VSDs are less than 5 mm in size and the Qp:Qs was less than 2:1. The availability of less invasive trans-catheter methods should not loosen up the indications for closure. Furthermore, the natural history studies clearly suggest that spontaneous VSD closures occur and such closures keep on occurring during childhood, adolescence, and adulthood. The pediatricians and the general pediatric cardiologists should not allow intervention by pediatric cardiac surgeons and interventional pediatric cardiologists for “small” VSDs that do not strictly fit the established criteria for closure, irrespective of the type of intervention.

## 7. Atrioventricular Septal Defect

Atrioventricular septal defects (AVSDs) have defects in atrial and ventricular septae along with deficiency in one or both AV valves. These defects were previously known as AV canals and endocardial cushion defects. The prevalence of AVSDs among all CHDs is 4–5% [7,12]. The AVSD is the most common defect in Down syndrome babies. The AVSDs may be classified into partial, transitional, intermediate, and complete types. The partial type has a large ASD in the anterior portion of the lower part of the atrial septum as well a cleft in the anterior leaflet of the mitral valve. These are also called ostium primum ASDs. In the transitional type, a small inlet VSD is also present and the physiology is similar to that of the partial form. The treatment of ostium primum ASDs and transitional AVSDs is similar and was discussed in the preceding section. The complete type has one AV valve annulus, a large ostium primum ASD, and an adjoining large inlet VSD. The intermediate type is similar to the complete form although the AV annulus is divided into two orifices by a tongue of tissue. The complete forms are further classified on the basis of relative ventricular sizes; balanced and unbalanced AVSDs. The prevalence of unbalanced AVSDs is 10–15% of AVSDs. The unbalanced AVSDs are classified into LV-dominant (large LV and small RV) and RV-dominant (large RV and small LV) types. The RV-dominant AVSDs occur more frequently than LV dominant forms. It should be noted that the unbalanced AVSDs may need different type of surgical strategy.

Hemodynamic abnormalities in these defects are related to the left-to-right shunt across the ASD and VSD components and atrioventricular valve insufficiency. Since the ASD and VSD components are usually large, and dilatation of the right atrium, RV, and main and branch PAs ensues. In the presence of moderate to severe mitral insufficiency, dilatation of the LA and LV occurs. Development of PVOD was noted as early as six months to one year of age in babies with complete AVSDs and even sooner in infants with Down syndrome.

The management of balanced and unbalanced AVSDs will be discussed separately.

### 7.1. Balanced AVSDs

#### 7.1.1. Indications for Intervention

Complete and intermediate types of AVSD usually have large ASD and VSD components and consequently all AVSDs require intervention. Complete repair is performed at most institutions and pulmonary artery banding is considered for infants weighing less than 5 kg in whom the CHF could not be controlled or the infant has additional significant co-morbidities.

In children with increased PVR, the deliberations for suitability for surgery and study of pulmonary vascular reactivity are the same to those portrayed in the section on “Ventricular Septal Defect” above. Patients with severely increased PVR, i.e., irreversible PVOD, are not candidates for AVSD repair. These patients may later be considered for lung transplantation.

#### 7.1.2. Timing of Intervention

AVSD repair should be performed prior to 6–12 months of age so as to prevent the development of irreversible PVOD. It is generally thought that babies with Down syndrome develop PVOD sooner than non-Down babies. Hypoplasia of pulmonary alveoli and capillaries [87] and chronic upper airway obstruction producing hypoventilation with consequent hypoxia and hypercarbia [88] may, in part, be responsible for higher PVR and early development of PVOD in infants with Down syndrome.

#### 7.1.3. Type of Intervention

*CHF.* In babies with CHF, the treatment is similar to that described in the VSD section above.

*Surgical Correction.* Surgery consisting of closure of atrial and ventricular septal defects with a patch, as well as repair and reconstruction of AV valves under cardiopulmonary bypass, is the procedure of choice. Both single-patch (pericardial) or two-patch (pericardial patch for primum ASD closure and pericardial or Dacron patch for the VSD) techniques have been used to close the ASD and VSD components of the defect. Comparison of the single-patch with the two-patch technique showed no significant difference between two groups with comparable outcomes [89]. If there are any associated defects, such as patent ductus arteriosus (PDA) and LV outflow tract obstruction, they should also be repaired at the same time. TEE to evaluate for mitral/tricuspid insufficiency and stenosis should be performed and, if detected, they should be addressed prior to decannulation from cardiopulmonary bypass.

*Pulmonary Artery Banding.* Banding of the PA is usually performed in infants weighing less than 5 kg with difficulty to control CHF or when significant co-morbidities co-exist. Complete surgical correction along with the removal of the previously placed PA band is performed later when the babies get better clinically. However, the current trend at most institutions is complete repair.

### 7.2. Unbalanced AVSDs

Management of CHF is similar to that described for VSDs and balanced AVSDs. Surgical options for treating unbalanced AVSDs are single-ventricle palliation (Fontan), biventricular repair along with bidirectional Glenn procedure, two-ventricle (TV) repair, and conversion from single-ventricle to TV repair [16].

#### 7.2.1. Single-Ventricle Palliation (Fontan)

The techniques used to repair of balanced AVSDs described above are not suitable for unbalanced AVSDs because hypoplastic RVs or LVs may not be capable of maintaining the pulmonary or systemic circulations, respectively, following the repair. Single-ventricle palliation is the most commonly used method in managing the unbalanced AVSDs. The single ventricle approach consists of three stages [16,90].

*Stage I.* At the time of presentation, usually in early infancy, pulmonary artery banding is performed to control CHF, to restrict the pulmonary blood flow, and to reduce the pulmonary artery pressures.

*Stage II.* At approximately six months of age, a bidirectional Glenn shunt is undertaken. In this procedure, the superior vena cava (SVC) is disconnected from the right atrium and anastomosed to the PA so that the blood from the SVC is directed into both branch PAs, thus, the name bidirectional Glenn. In patients with an additional persistent left SVC, a bilateral, bidirectional Glenn procedure is performed especially if the bridging left innominate vein is small or absent. Prior to the bidirectional Glenn procedure, normal pulmonary artery pressures should be documented either by echo-Doppler or cardiac catheterization studies. In cases with significant AV valve regurgitation, the AV valve should also be repaired at the time of bidirectional Glenn procedure. If there are any other hemodynamically important abnormalities, they should also be taken care of at this time.

*Stage IIIA.* Between the ages of one and four years, usually one year following the bidirectional Glenn, Fontan conversion is performed by redirecting the inferior vena caval flow into the PA by either a lateral tunnel or an extra-cardiac non-valved conduit. Extra-cardiac conduit with fenestration is preferred by most surgeons. Cardiac catheterization is usually performed prior to this surgery to assess pulmonary artery anatomy and pressures, PVR, left ventricular end-diastolic pressure, and trans-pulmonary gradient, and to make sure of their normalcy prior to proceeding with Fontan conversion.

*Stage IIIB.* The fenestration may be closed if clinically indicated, by trans-catheter device implantation (Figure 11) six months to one year after stage IIIA.

#### 7.2.2. Biventricular Repair Along with Bidirectional Glenn Procedure

In patients with marginal-sized RVs, a combination of repair similar to that described for balanced AVSDs (biventricular repair) and a bidirectional Glenn procedure is performed. The bidirectional Glenn reduces the amount of blood flow that needs to be pumped by the RV.

#### 7.2.3. Biventricular Repair

In biventricular repair, the AV valves are repaired to construct nearly similar-sized AV valves and the VSD is closed with a patch (the patch is moved to the right in RV-dominant AVSDs) to create nearly equal-sized ventricles. In single-stage biventricular repair, the ASD and VSD components are completely closed at the same time [91]. In staged biventricular repair, ASD and VSD are partially closed at the time of the first operation and complete closure of the ASD and VSD is undertaken at the time of a second operation, once the growth of the hypoplastic ventricle is demonstrated. The principles used to select a given patient to single ventricle versus TV repair are not clearly understood. A number parameters have been proposed for the selection of patients for TV repair which include indexed VSD, AV valve index, LV-to-RV long axis ratio, and RV/LV inflow angle in systole, as reviewed in greater detail in the author’s prior publication [16].

#### 7.2.4. Conversion from Single-Ventricle to Two-Ventricle Repair

Given the suboptimal long-term outcomes after single-ventricle palliation, conversion from single-ventricle to TV repair was proposed [92,93]. The LV is rehabilitated by encouraging flow through the LV by relieving inflow and outflow tract obstructions and by restriction of the atrial septal defect to promote flow through the LV. Resection of endocardial fibroelastosis, if present, is also performed.

The results of surgical repair of balanced and unbalanced (single-ventricle palliation, TV repair, and conversion from single-ventricle to TV repair) AVSDs were reviewed elsewhere [16] for the interested reader.

## 8. Patent Ductus Arteriosus

Patent ductus arteriosus (PDA) is a muscular structure that connects the main PA at its junction with the left PA to the descending thoracic aorta at the level of left subclavian artery. Isolated PDA constitutes 6–11% of all CHDs [7,12]. In the fetus, the ductus arteriosus diverts the deoxygenated blood from the PA into the descending aorta [70,71,72,94]. Following birth, the ductus arteriosus constricts and spontaneously closes, largely due to increased PO_2_, among other factors [70,71,72,94]. However, in some infants, such spontaneous closure does not occur and persistence patency of the ductus beyond 72 h of life is defined as PDA. Such ductal persistence is more frequent in premature babies than in full-term infants. PDA may be an isolated lesion and may be present in association with other defects. The shape of PDA varies considerably but most often it is conical- or funnel-shaped. The wider aortic end (ampulla) gradually narrows towards the pulmonary end. Most often the narrowest segment is at the pulmonary end and is designated as minimal ductal diameter. Other types such as short and tubular and those with multiple constrictions and bizarre configuration are also seen. A classification on the basis of the shape of the ductus and the relationship of the narrowest part of PDA with trachea was presented by Kriechenko and his colleagues [95] and is used by most cardiologists. Since the pressure and resistance in the systemic circuit are higher than those in the pulmonary circuit, left-to-right shunt takes place across the ductus. The amount of left-to-right shunting depends upon the minimal diameter of the ductus and ratio of pulmonary to systemic vascular resistance. The left-to-right shunt across the PDA produces dilatation of the LA and LV. Main and branch PAs also dilate. In premature babies, excessive blood flow into the lungs may cause pulmonary edema and decreased lung compliance and may even result in respiratory failure and chronic lung disease [96].

### 8.1. Indications for Intervention

It is generally thought that the presence of an isolated ductus is an indication for closure, mostly to prevent SBE [97,98,99,100,101,102]. The closure procedure is indicated only in patients with continuous murmur of PDA with echo-Doppler confirmation. The so-called “silent ductus” detected incidentally without typical auscultatory findings need not be closed [97]. Very small and small PDAs should be closed even if they are not hemodynamically significant, largely to eliminate the risk of SBE. The Qp:Qs criteria alluded to in the ASD and VSD sections above are not applicable to PDA patients. Medium- and large-sized PDAs should be closed to avoid increasing LV volume overload, to treat CHF, and to prevent PVOD in addition to eliminating the risk of SBE. Closure should not be performed in patients with ductal-dependent CHD and in patients who developed PVOD.

### 8.2. Timing of Intervention

PDA closure can be performed at any time, especially if associated with heart failure or pulmonary compromise. However, in patients who are asymptomatic, waiting until 6–12 months of age is suggested because of the lower risk profile at this age. Large PDAs should be closed prior to 18 months of age to prevent development of PVOD. In premature babies with hemodynamically significant PDAs producing deterioration of pulmonary function, closure may be undertaken in premature infants at a very young age [96].

### 8.3. Type of Intervention

Surgical ligation/division [1], video-assisted thoracoscopic surgical (VATS) interruption [103,104], and trans-catheter occlusion [97,98,99,100,101,102] of PDAs are currently available options to address the PDAs.

*Surgery.* Since the first report of successful ligation of PDA by Gross and Hubbard [1], surgery has been the treatment of choice for PDA until trans-catheter closure methods were developed. At the present time surgery is used for limited indications and these include premature babies, to address complications of trans-catheter occlusion, very large PDAs deemed unsuitable for trans-catheter occlusion, and concurrent with other cardiac surgical procedures.

*Video-Assisted Thoracoscopic Surgical Interruption.* The VATS technique for interruption of PDA was initially described by Laborde and colleagues [103] and subsequently modified by them and others [104]. Although VATS has been in use since the early 1990s, the procedure is available only at a limited number of institutions. It is particularly useful in addressing PDAs in small infants.

*Trans-Catheter Occlusion.* Since the description of device occlusion of PDAs by Porstman [105], Rashkind [106], and their colleagues in the 1960s and 1970s, a number of devices have been designed and experimented, as reviewed elsewhere [107,108,109]. However, only a few devices are approved by the United States FDA and these include the Gianturco coil (free and detachable), Gianturco-Grifka vascular occlusion device (GGVOD) (Cook, Bloomington, IN, USA), Amplatzer duct occlude (ADO) (St. Jude Medical, Inc., Golden Valley, MN, USA), and more recently ADO II and Nit-Occlud. At the current time, Gianturco coils (Cook, Bloomington, IN, USA) for closure of very small and small PDAs (Figure 12) and the Amplatzer duct occlude (St. Jude Medical, Inc., Golden Valley, MN, USA) (Figure 13) for occlusion of moderate and large PDAs are utilized with good results. Detailed description of the procedure of implantation of the devices and results of the procedure are beyond the scope of this presentation and the interested reader is referred elsewhere [98,99,100,101,102,107,110].

### 8.4. PDA in the Premature

When hemodynamically significant PDA, causing a worsening respiratory status or failing to progress in efforts to wean off respiratory support is present in the premature babies, PDA closure is indicated [96]. Initially, conservative management with fluid restriction, diuretic therapy and respiratory support, as needed, is given. If no improvement is detected, pharmacologic therapy with indomethacin or ibuprofen (oral or intravenous) to close the ductus should be started. Ibuprofen is preferred by some neonatologists because of less renal toxicity. Failure after two courses of pharmacologic therapy is an indication for surgical (conventional or bedside), VATS or trans-catheter closure of the ductus depending upon the expertise at a given institution [96].

## Figures and Tables

**Figure 1 children-06-00042-f001:**
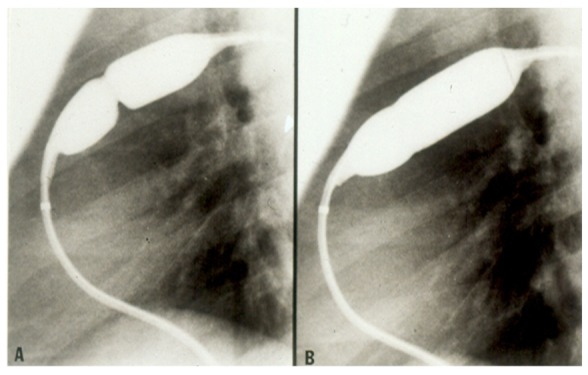
Selected cine frames in straight lateral projection illustrating a balloon dilatation catheter across the stenosed pulmonary valve. Waisting of the balloon was seen during the early phases of inflation of the balloon (**A**) which was nearly abolished on further inflation of the balloon (**B**). Reproduced from Rao, P. S. Transcatheter Therapy in Pediatric Cardiology. Wiley-Liss, Inc., New York, 1993; p. 62.

**Figure 2 children-06-00042-f002:**
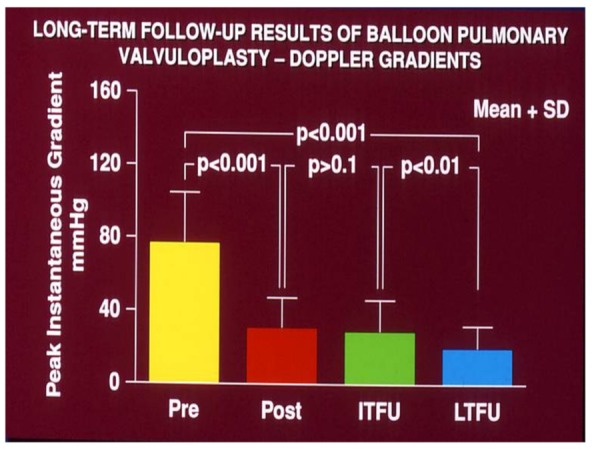
Bar graph demonstrating maximum peak instantaneous Doppler gradients prior to (Pre), one day following (Post) balloon pulmonary valvuloplasty, and at intermediate-term (ITFU) and long-term (LTFU) follow-up. Note the significant reduction (*p* < 0.001) after valvuloplasty which remains unchanged (*p* > 0.1) at ITFU. However, at LTFU there was a further fall (*p* < 0.001) in the Doppler gradients. Mean + standard deviation (SD) are shown. Reproduced with the permission of the author and publisher from Rao, P.S. et al. Heart 1998; 80:591–5.

**Figure 3 children-06-00042-f003:**
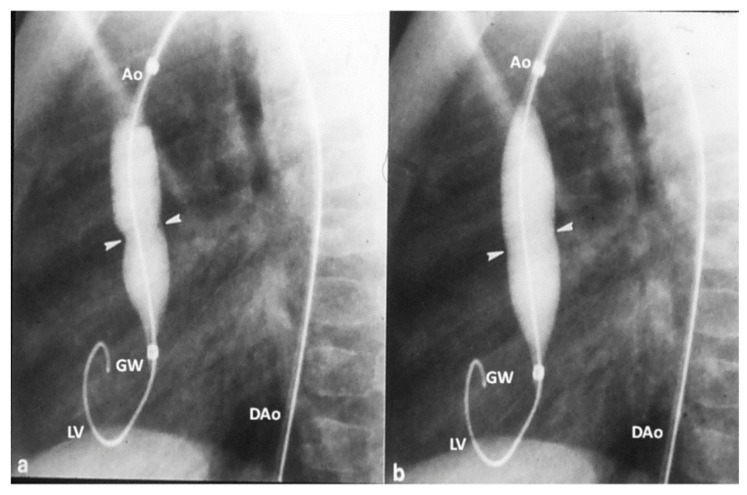
Selected cine frames in straight lateral projection illustrating a balloon dilatation catheter across the stenosed aortic valve. Waisting of the balloon (filled arrows) was seen during the early phases of inflation of the balloon (**a**) which was completely abolished on further inflation of the balloon (**b**). Ao, aorta; DAo, descending aorta; GW, guide wire; LV, left ventricle. Reproduced from Rao PS (Editor). Congenital Heart Disease—Selected Aspects, InTech, Rijeka, Croatia, 2012, pp. 3–44.

**Figure 4 children-06-00042-f004:**
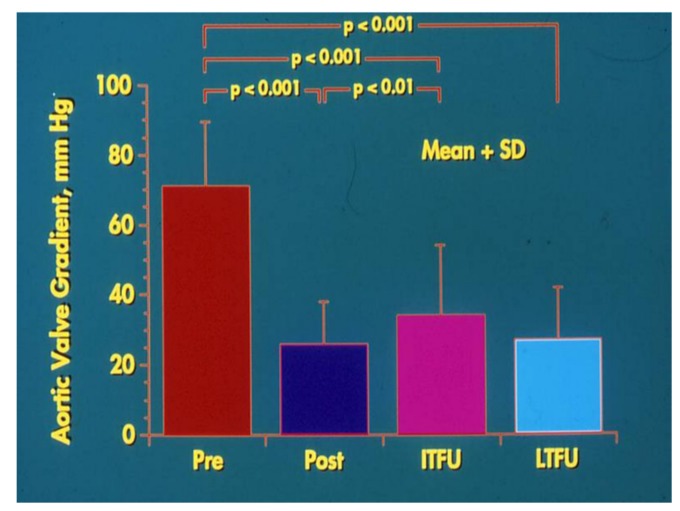
Bar graph demonstrating maximum peak instantaneous Doppler gradients prior to (Pre), one day following (Post) balloon aortic valvuloplasty, and at intermediate-term (ITFU) and long-term (LTFU) follow-up. Note the significant reduction (*p* < 0.001) after valvuloplasty. The residual gradients continue to be lower (*p* < 0.001) at ITFU and at LTFU. Mean + standard deviation (SD) are shown.

**Figure 5 children-06-00042-f005:**
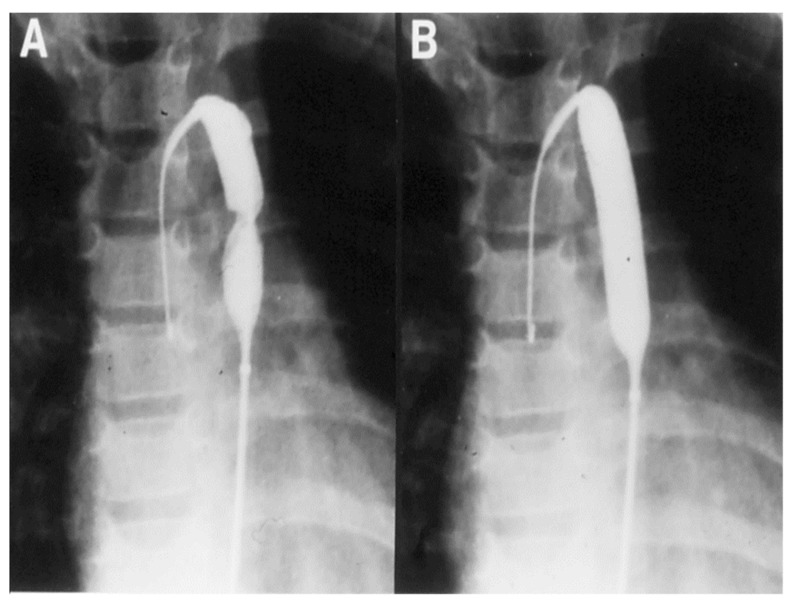
Selected cineflurograpic frames in posterio-anterior projection in a child demonstrating an angioplasty balloon across the aortic coarctation with waisting of the balloon (**A**) during the initial phases of balloon inflation; the waist has completely disappeared (**B**) with further balloon inflation. The guide wire (GW) is positioned in the ascending aorta (AAo). Reproduced from Rao PS. Balloon angioplasty of native aortic coarctation. In: Rao, P. S (ed): Transcatheter Therapy in Pediatric Cardiology, Wiley-Liss, New York, NY, 1993:153–196.

**Figure 6 children-06-00042-f006:**
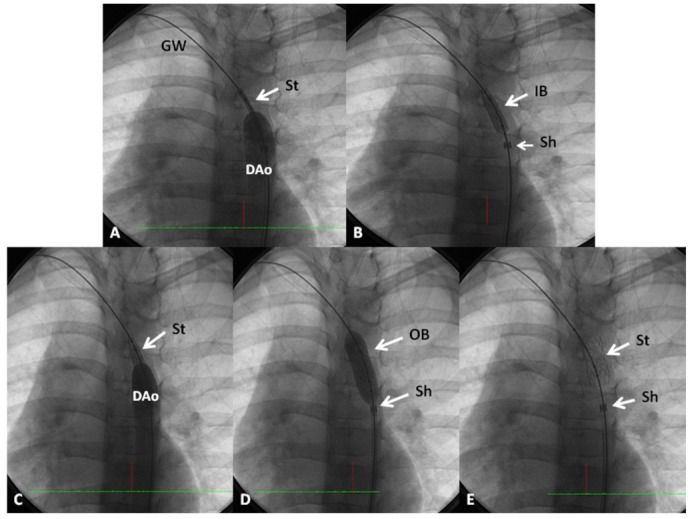
Selected cine-angiographic/radiographic frames demonstrate stent deployment in aortic coarctation using a balloon-in-balloon catheter showing the position of an un-inflated stent (**A**), following inflation of the inner balloon (IB) (**B**), test angiogram after inflation of inner balloon (**C**), after outer balloon (OB) inflation (**D**), and after balloon deflation and withdrawal (**E**). The white arrow in (E) demonstrates the expanded stent (St). The guide wire (GW) is positioned in the right subclavian artery in an attempt to keep the stent straight. Radio-opaque tip of the sheath (Sh) is seen. DAo, descending aorta. Reproduced from Doshi, A. R.; Rao, P. S. Pediatr Therapeut 2012; S5:006. doi: 10.4172/2161-0665.S5-006.

**Figure 7 children-06-00042-f007:**
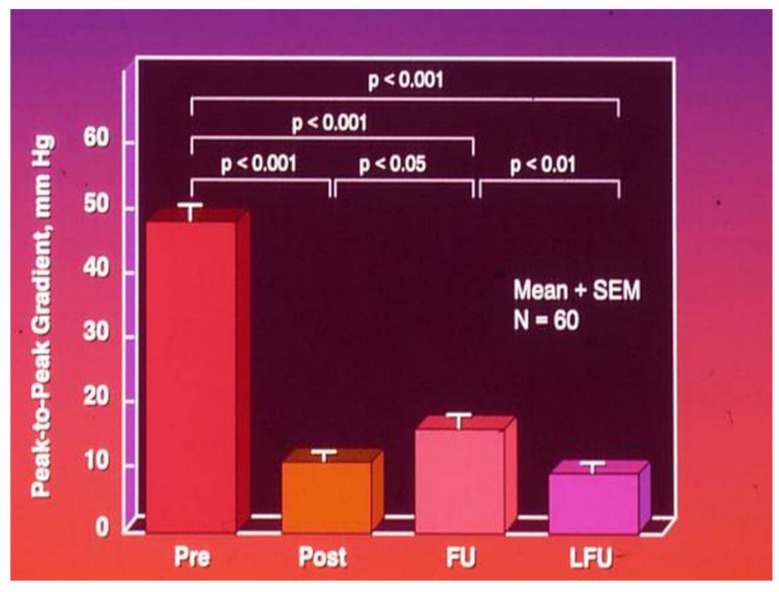
Bar graph of results following balloon angioplasty of native aortic coarctation are shown. Peak-to peak systolic pressure gradients across the coarctation decreased significantly (*p* < 0.001) from prior to (Pre) to immediately after (Post) balloon angioplasty. However, the gradient increased (*p* < 0.05) slightly at intermediate-term follow-up (FU). However, these gradients continue to be lower (*p* < 0001) than pre-angioplasty values. At long-term follow-up (LFU) arm-leg peak systolic pressure difference, measured by blood pressures, is lower than coarctation gradients prior to (*p*< 0.001) and at intermediate-term follow-up (*p* < 0.01). Mean + SEM (standard error of the mean) are shown. *N*, number of patients undergoing balloon angioplasty. Modified from Rao, P. S.; et al. J Am Coll Cardiol 1996; 27:462–70.

**Figure 8 children-06-00042-f008:**
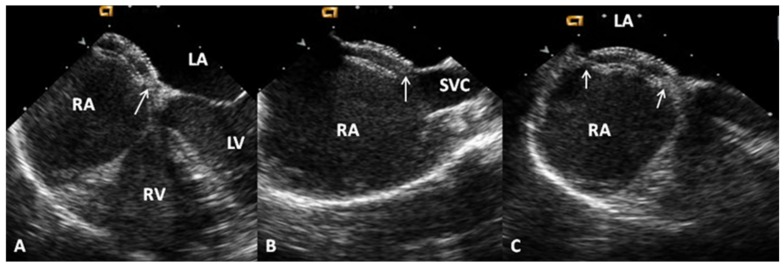
Selected video frames from a trans-esophageal echocardiogram following implantation of Amplatzer Septal Occluder to occlude an atrial septal defect demonstrating the position of both discs in four chamber (**A**), bi-caval (**B**), and long axis (**C**) views. Note that the rims of the defect (thin arrows) are sandwiched between left atrial (LA) and right atrial (RA) discs. LV, left ventricle; RV, right ventricle; SVC, superior vena cava. Reproduced from Rao, P. S. Non-surgical closure of atrial septal defects in children. In: Atrial and Ventricular Septal Defects: Molecular Determinants, Impact of Environmental Factors and Non-Surgical Interventions, Larkin, S. A. (Ed), ISBN: 978-1-62618-326-1. Nova Science Publishers, Inc.

**Figure 9 children-06-00042-f009:**
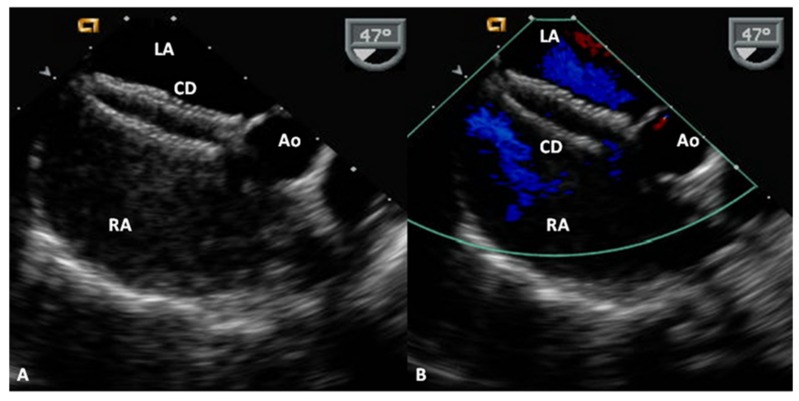
Selected video frames from a trans-esophageal echocardiogram following the delivery of Amplatzer cribriform device (CD) demonstrating the position of both discs across the atrial septum (**A**) without any residual shunt (**B**) in a patient with a fenestrated atrial septal defect. Ao, aorta; LA, left atrium; RA, right atrium. Reproduced from Rao, P. S. Non-surgical closure of atrial septal defects in children. In: Atrial and Ventricular Septal Defects: Molecular Determinants, Impact of Environmental Factors and Non-Surgical Interventions, Larkin, S. A. (Ed), ISBN: 978-1-62618-326-1. Nova Science Publishers, Inc.

**Figure 10 children-06-00042-f010:**
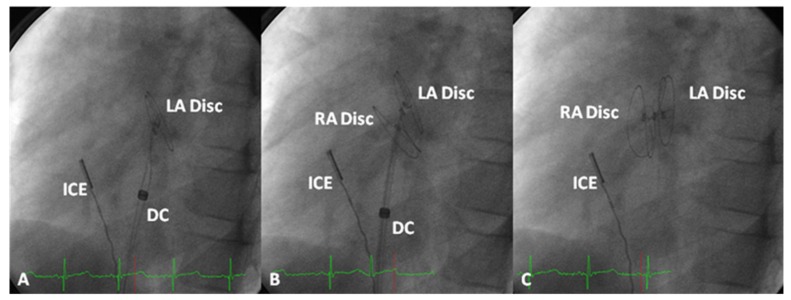
Selected cine frames from cine-flurograms in a 60° left anterior oblique view demonstrating the position of the HELEX device following the delivery of the left atrial (LA) and right atrial (RA) discs and after disconnecting the delivery catheter (DC), respectively in (**A**), (**B**), and (**C**). Intra-cardiac echocardiography catheter (ICE) is seen which was used to monitor device implantation. Reproduced from Rao, P. S. Non-surgical closure of atrial septal defects in children. In: Atrial and Ventricular Septal Defects: Molecular Determinants, Impact of Environmental Factors and Non-Surgical Interventions, Larkin, S. A. (Ed), ISBN: 978-1-62618-326-1. Nova Science Publishers, Inc.

**Figure 11 children-06-00042-f011:**
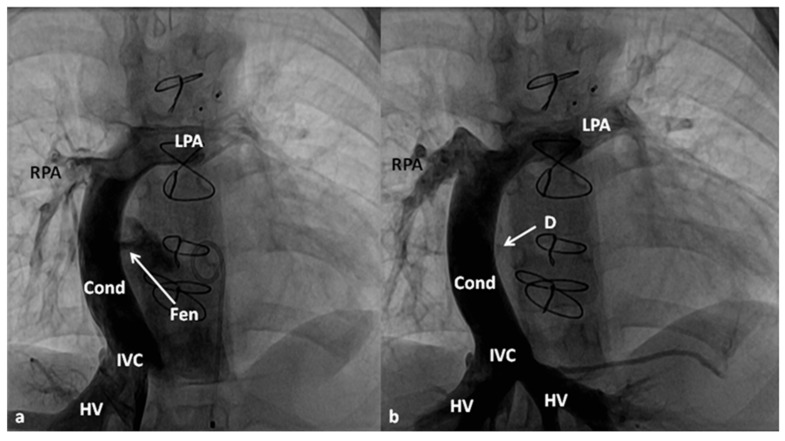
Selected cine frames in antero-posterior view demonstrating Fontan fenestration (**a**) and its closure (**b**) with an Amplatzer device (D) (Stage IIIB Fontan). Note complete closure in (b) without any residual shunt. HV, hepatic veins; LPA, left pulmonary artery; PG, pigtail catheter in the descending aorta; RPA, right pulmonary artery. Modified from Rao, P. S. Indian J Pediatr 2015; 82:1147–56.

**Figure 12 children-06-00042-f012:**
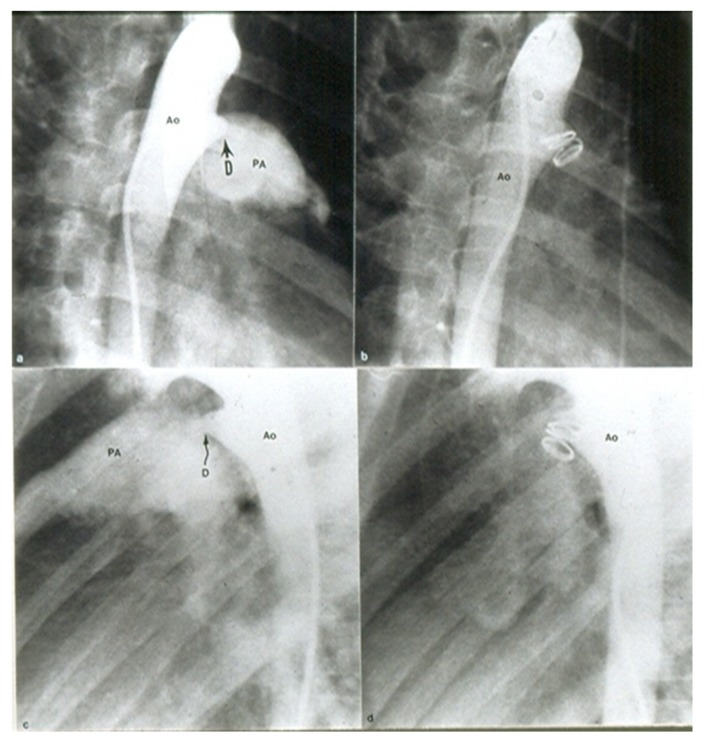
Selected cine frames from aortic arch angiograms in 30° right anterior oblique (**a**,**b**) and straight lateral (**c**,**d**) views showing a small to moderate sized ductus (D) before (**a**,**c**) and 15 min following (**b**,**d**) coil occlusion. Note complete occlusion of the ductus in (**b**,**d**). Ao, aorta; PA, pulmonary artery. Reproduced from Rao, P. S.; et al. Am J Cardiol 1997; 80:1498–1501.

**Figure 13 children-06-00042-f013:**
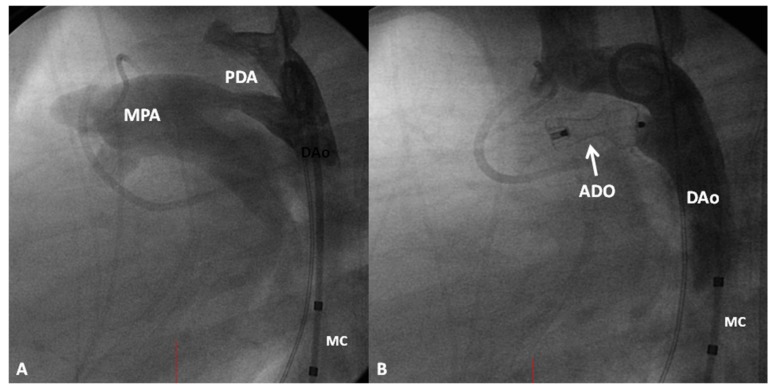
(**A**). Selected cine frame from aortic arch angiogram in lateral view demonstrating a moderate-sized patent ductus arteriosus (PDA) opacifying the main pulmonary artery (MPA). (**B**). Following implantation of an Amplatzer duct occluder (ADO), no residual shunt is seen. Additionally, there is no descending aortic (DAo) obstruction. MC, Marker pigtail catheter.

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
