# Peer review of "Management of Congenital Heart Disease: State of the Art; Part I—ACYANOTIC Heart Defects"

_children, 2019, doi:10.3390/children6030042_

Round 1
Reviewer 1 Report
This is a review article outlining the indications and treatment of acyanotic congenital heart disease. The author has done a very good job outlining the indications and the treatment of acyanotic congenital heart disease. The references are appropriate and the pertinent to each section. The treatment modality for each acyanotic defect is current. This is a review article therefore, there are no original contributions in this manuscript. This paper would serve as a summary for the care of patients with acyanotic heart disease. I would have liked to see a little more discussion on when to overdrive pace for aortic valves and coarctation of the aorta. The use of a single or double balloon valvuloplasty in the aortic and pulmonary valve. A little more explanation on the indications for critical and aortic and pulmonary valvuloplasty in the neonate. Similarly , a little more information on the use of a high or low pressure balloon for the arteries. Also the use of covered vs non covered stents in the treatment of coarctation of the aorta would be helpful. The article is very well written.Author Response
Thanks to the reviewer for the positive comments pertaining to the manuscript.
Discussion on when to overdrive pace for aortic valves and coarctation of the aorta was included, as suggested by the reviewer.
The use of a single or double balloon valvuloplasty in the aortic and pulmonary valve was also incorporated into text of the script.
Discussion of high or low pressure balloon for the arteries asked for by the reviewer could not be included because trans-catheter management of branch pulmonary arteries, by design, was not included in this review.
A brief statement re covered vs non covered stents in the treatment of coarctation of the aorta was already included.
Reviewer 2 Report
This is an excellent review. A few minor suggestions.
Would you be able to add the incidences with appropriate references for each lesion. That may be helpful for readers as well.
I note that some figures are reproduced. However others look old but do not say they are reproduced. If those are reproduced, I would add that for each figure it applies for.
Figure 1. Change “completely” to “nearly”
TAVR is briefly mentioned. I would also mention in the section on pulmonary stenosis that PR can be an seen in follow-up and may require surgery or transcatheter pulmonary valve implantation in the future (with references).
I would also mention that after intervention/surgery for AS, patients may need to be followed for AR and recurrent AS and require aortic valve replacement in the future.
In the discussion for PFOs 5.5.5 add Gore Cardioform Septal Occluder (W.L. Gore and Associates) which is also FDA approved and mentioned for both Amplatzer and Gore devices the stroke approval indication specifics.
In the discussion for 7.2.1, for stage IIIB, change “is closed” to “may be closed if clinically indicated”
Author Response
This is an excellent review. A few minor suggestions.
Thanks
Would you be able to add the incidences with appropriate references for each lesion. That may be helpful for readers as well.
The incidences for each lesion are added, as suggested.
I note that some figures are reproduced. However others look old but do not say they are reproduced. If those are reproduced, I would add that for each figure it applies for.
For figures that are reproduced, it is clearly indicated. Figures that are previously published, no such notation was indicated.
Figure 1. Change “completely” to “nearly”
Caption for Figure 1 “completely” was changed to “nearly”, as recommended
TAVR is briefly mentioned. I would also mention in the section on pulmonary stenosis that PR can be an seen in follow-up and may require surgery or transcatheter pulmonary valve implantation in the future (with references).
Development of PR at follow-up and that it may require surgery or trans-catheter pulmonary valve implantation in the future was included, as suggested .
I would also mention that after intervention/surgery for AS, patients may need to be followed for AR and recurrent AS and require aortic valve replacement in the future.
Development of AI and recurrent AS in follow-up and that it may require surgery in the future (with references) was included, as suggested.
In the discussion for PFOs 5.5.5 add Gore Cardioform Septal Occluder (W.L. Gore and Associates) which is also FDA approved and mentioned for both Amplatzer and Gore devices the stroke approval indication specifics.
Added Gore Cardioform Septal Occluder (W.L. Gore and Associates), Amplatzer, and Gore devices as FDA-approved indications for stroke, as recommended.
In the discussion for 7.2.1, for stage IIIB, change “is closed” to “may be closed if clinically indicated”
In the discussion for 7.2.1, for stage IIIB, “is closed” was changed to “may be closed if clinically indicated”, as suggested